# Older adults with non-communicable chronic conditions and their health care access amid COVID-19 pandemic in Bangladesh: Findings from a cross-sectional study

Sabuj Kanti Mistry[1,2,3]*, A. R. M. Mehrab Ali[1,4], Uday Narayan Yadav[2,6], Saruna Ghimire[5], Md. Belal Hossain[3,7], Suvasish Das Shuvo[8], Manika Saha[9], Sneha Sarwar[10], Md. Mohibur Hossain Nirob[11,12], Varalakshmi Chandra Sekaran[13], Mark F. Harris[2]

1 ARCED Foundation, Dhaka, Bangladesh, 2 Centre for Primary Health Care and Equity, University of New South Wales, Sydney, Australia, 3 BRAC James P Grant School of Public Health, BRAC University, Dhaka, Bangladesh, 4 Innovations for Poverty Action, New Haven, Connecticut, United States of America, 5 Department of Sociology and Gerontology and Scripps Gerontology Center, Miami University, Oxford, OH, United States of America, 6 Center for Research Policy and Implementation, Biratnagar, Nepal, 7 School of Population and Public Health, University of British Columbia, Vancouver, Canada, 8 Department of Nutrition and Food Technology, Jashore University of Science and Technology, Jashore, Bangladesh, 9 Action Lab, Department of Human-Centred Computing, Faculty of Information Technology, Monash University, Melbourne, Australia, 10 Institute of Nutrition and Food Science, University of Dhaka, Dhaka, Bangladesh, 11 Bangabandhu Sheikh Mujib Medical University (BSMMU), Dhaka, Bangladesh, 12 Directorate General of Health Services, Ministry of Health and Family Welfare, Dhaka, Bangladesh, 13 Department of Community Medicine, Melaka Manipal Medical College (Manipal Campus) MAHE, Manipal, Karnataka, India

* smitra411@gmail.com

**Data Availability Statement:** All relevant data are within the paper and its Supporting Information files.

## Abstract

### Background

Burgeoning burden of non-communicable disease among older adults is one of the emerging public health problems. In the COVID-19 pandemic, health services in low- and middle-income countries, including Bangladesh, have been disrupted. This may have posed challenges for older adults with non-communicable chronic conditions in accessing essential health care services in the current pandemic. The present study aimed at exploring the challenges experienced by older Bangladeshi adults with non-communicable chronic conditions in receiving regular health care services during the COVID-19 pandemic.

### Materials and methods

The study followed a cross-sectional design and was conducted among 1032 Bangladeshi older adults aged 60 years and above during October 2020 through telephone interviews. Self-reported information on nine non-communicable chronic conditions (osteoarthritis, hypertension, heart disease, stroke, hypercholesterolemia, diabetes, chronic respiratory diseases, chronic kidney disease, cancer) was collected. Participants were asked if they faced any difficulties in accessing medicine and receiving routine medical care for their medical conditions during the COVID-19 pandemic. The association between non-

**Funding:** The author(s) received no specific funding for this work.

**Competing interests:** The authors have declared that no competing interests exist.

communicable chronic conditions and accessing medication and health care was analysed using binary logic regression model.

## Results

Most of the participants aged 60–69 years (77.8%), male (65.5%), married (81.4%), had no formal schooling (58.3%) and resided in rural areas (73.9%). Although more than half of the participants (58.9%) reported having a single condition, nearly one-quarter (22.9%) had multimorbidity. About a quarter of the participants reported difficulties accessing medicine (23%) and receiving routine medical care (27%) during the pandemic, and this was significantly higher among those suffering from multimorbidity. In the adjusted analyses, participants with at least one condition (AOR: 1.95, 95% CI: 1.33–2.85) and with multimorbidity (AOR: 4.75, 95% CI: 3.17–7.10) had a higher likelihood of experiencing difficulties accessing medicine. Similarly, participants with at least one condition (AOR: 3.08, 95% CI: 2.11–4.89) and with multimorbidity (AOR: 6.34, 95% CI: 4.03–9.05) were significantly more likely to face difficulties receiving routine medical care during the COVID-19 pandemic.

## Conclusions

Our study found that a sizeable proportion of the older adults had difficulties in accessing medicine and receiving routine medical care during the pandemic. The study findings highlight the need to develop an appropriate health care delivery pathway and strategies to maintain essential health services during any emergencies and beyond. We also argue the need to prioritise the health of older adults with non-communicable chronic conditions in the centre of any emergency response plan and policies of Bangladesh.

## Introduction

The world is now facing major challenges imposed by the COVID-19 pandemic (WHO) [1]. As of 14th February 2021, there are over 108 million confirmed cases and 2.4 million deaths worldwide and 540,266 confirmed cases with 8,266 deaths [2] in Bangladesh. The nationwide lockdown measures implemented to curb the transmission by many countries, including Bangladesh, limited general people's access to health care and other needs. Furthermore, the temporary shutdown of public transportation meant that people who relied on those services to visit health facilities were prevented from accessing general and emergency care. Notably, in Bangladesh, most people rely on public transit [3].

The older population, who are at greater risk, has been especially hard hit by the pandemic [4]. More than half of the COVID-19 related deaths in China, Italy [5], and India [6] and almost 39% in Bangladesh [7] were among the older population. The presence of pre-existing non-communicable conditions such as diabetes, hypertension, and obesity, which are highly correlated with age, contribute to their increased vulnerability [8–10].

Non-communicable diseases (NCDs) account for 70% of global deaths and 80% of deaths in Low- and Middle- Income Countries (LMICs) [11]. Multimorbidity, defined as the coexistence of two or more non-communicable chronic conditions, [12] has increased both globally [13] and in Bangladesh [14]. The prevalence of multimorbidity in the community settings of LMICs ranges from 7.8% to 29.7% [15, 16]. Similar estimates from Bangladesh range from

8.4% to 53.8% [14, 17]. The proportion of undiagnosed chronic conditions is high in LMICs [18], including Bangladesh [14, 19]. From this, we can infer that the actual burden of single and multiple chronic conditions is underestimated and is projected to increase in the future with population aging and epidemiological transition. Multimorbidity is associated with reduced quality of life [20, 21] and increased healthcare dependence, increased care needs, and spending [19, 22]. People with multimorbid conditions require a multitude of specialist services and treatment plans and care is currently fragmented in countries such as Bangladesh [23].

Access to healthcare is a constitutional right in Bangladesh, and the government has implemented Universal Health Coverage [24]. Routine medical check-ups, adherence to the treatment regimes, and healthy lifestyles are key strategies for managing non-communicable chronic conditions. During COVID-19 people with NCDs have been reported to be not receiving promotive, preventive, and clinical care, has and this has exacerbated their chronic condition [25–27]. COVID-19 appears to be operating as a syndemic aggravating existing conditions and degrading people's quality of life. Due to the existing inequalities, people with one or more NCDs conditions appear to have received insufficient health care services during this pandemic in Bangladesh [28, 29].

However, to date, there is no evidence in Bangladesh regarding the extent to which COVID-19 has impacted the receipt of regular health services by older people with NCDs. Therefore, the current study aimed at estimating the challenges experienced by older Bangladeshi adults in receiving routine health care services, including medication access and routine medical treatment during the COVID-19 pandemic. This study's findings may enable the government, policymakers, and health practitioners to take appropriate actions to support these vulnerable groups during this pandemic.

## Materials and methods

### Study design and participants

A cross-sectional study was conducted remotely, through telephone interviews, in October 2020. A pre-existing registry developed by the authors' institute through merging the contact information of households from ten different community-based studies served as the sampling frame and included households from all eight administrative divisions of Bangladesh, both urban and rural areas, and different income groups. Assuming a 50% prevalence of the outcome with a 5% margin of error, at 95% confidence level, 90% power of the test, and 80% response rate, the study estimated that the required sample size was 1096. Although 1096 participants were approached, only 1032 eligible participants agreed to participate, resulting in a ~94% study response rate. To ensure representativeness from all eight divisions, probability proportionate to the number of older adults in each division was used [30]. In each administrative division, households were selected using a simple random sampling technique and subsequently, one eligible participant was interviewed from the selected household. Hence, the number of included households and respondents were equal. When the household had more than one eligible participant, the oldest member was nominated for an interview. The only inclusion criterion was defined in terms of age (i.e., $\geq$ 60 years). The exclusion criteria included severe mental conditions (clinically proved schizophrenia, bipolar mood disorder), a hearing disability, or inability to communicate.

### Measures

**Outcome measure.** This study's outcome variables were difficulties faced by the participants in getting medicine and in receiving routine medical care during the COVID-19

pandemic. The participants were asked the following questions: "How much difficulty do you have accessing the medicine that you need because of the COVID-19 pandemic or social distancing rules?" and "How much difficulty do you have with receiving routine medical care that you need because of the COVID-19 pandemic or social distancing rules?". The responses were dichotomized (1 = No difficulties and 2 = Difficulties faced).

**Explanatory variables.** Explanatory variables included administrative division (Barishal, Chattogram, Dhaka, Mymensingh, Khulna, Rajshahi, Rangpur, and Sylhet), age (dichotomized as 60–69 and ≥70 years), sex (male/female), marital status (married and widowed), formal schooling (yes/no), family size (≤4 and >4), family income in Bangladeshi taka (BDT) (<5,000, 5,000–10,000, >10,000), residence (urban/rural), current occupation (employed and unemployed), living arrangements (with family/live alone), walking proximity to the nearest health centre (<30 minutes/ ≥30 minutes).

The data on nine non-communicable chronic conditions (Hypertension, Heart diseases, Stroke, Hypercholesterolemia, Diabetes, Chronic respiratory diseases, Chronic kidney disease, Cancer, and Osteoarthritis) were collected through self-reports. For each of the conditions, participants were asked two questions: 1) "Has a doctor or other health professional ever told you that you have [the condition]?" and 2) "Are you currently taking any medications for [the condition]?" Participants were classified as having a given condition if they reported yes to either of the two questions. The definitions of the individual non-communicable condition, having at least one condition and multimorbidity are presented in Table 1.

## Data collection tools and techniques

A pre-tested semi-structured questionnaire in the Bengali language was used to collect the information through a telephone interview. Data collection was accomplished electronically using SurveyCTO mobile app (Version 2.70.6, SurveyCTO CATI feature, Dobility, Inc., Washington, DC, USA) (https://www.surveycto.com/) by ten research assistants recruited based on previous experience of administering a health survey on an electronic platform. The research assistants were trained extensively, through Zoom meeting (Version 5.4.2, Zoom Video Communications, Inc., San Jose, CA, USA), for four days before the data collection.

**Table 1. Definition of a single condition and multimorbidity.**

| Conditions | Definition |
|---|---|
| Hypertension | Self-reports of hypertension and/or taking antihypertensive medications |
| Heart diseases | Self-reports of heart attack, angina, or "heart trouble", and/or taking medications for heart diseases |
| Stroke | Self-reports of the previous stroke or taking medication for a recent episode of stroke |
| Hypercholesterolemia | Self-reports of taking medication for hypercholesterolemia |
| Diabetes | Self-reports of diabetes or taking insulin or antidiabetic medications |
| Chronic respiratory diseases | Self-reports of the conditions or taking medication for any chronic respiratory diseases |
| Chronic kidney disease | Self-reports of chronic kidney disease, taking medication for the disease or undergone dialysis |
| Cancer | Self-reports of cancer diagnosis, taking medication for cancer, or past or current cancer therapy, including chemotherapy and radiation therapy. |
| Osteoarthritis | Self-report of joint pain problems |
| At least one condition | The presence of at least one of the above nine conditions. |
| Multimorbidity | The presence of two or more of the above nine conditions. Multimorbidity, a binary outcome variable, was defined as non-multi-morbid (i.e., none or single conditions) and multi-morbid (i.e., two or more conditions). |

The English version of the questionnaire was first translated to Bengali language and then back-translated to English by two researchers to ensure its consistency. The Bengali version of the tool was piloted among a small sample (n = 10) of older adults through telephone interviews to refine the language of the final version which ensures face validity of the questionnaire. The final Bengali questionnaire was used for data collection.

## Statistical analyses

R version 4.0.3 was used for all analyses. Descriptive summaries (frequency and proportion) are present for the explanatory variables by outcome variables, and differences were assessed using the Chi-square test. We performed a multinomial logistic regression model with no condition as the base outcome to determine the association between having at least one condition and multimorbidity with difficulties receiving medicine and routine medical care. A separate multivariable model was fitted for each outcome variable. Each was adjusted for age, sex, marital status, formal schooling, family size and income, residence, current occupation, living arrangement, proximity to the nearest health centre. Crude and adjusted odds ratio (OR) and 95% confidence interval (95% CI) are reported.

The sensitivity of our logistic regression results was assessed using the propensity score approach as an alternative confounding adjustment tool. We used the propensity score weighting to reweights both unexposed (no-condition of non-communicable disease) and exposed (at least one condition and multimorbidity) to emulate a propensity score weighted population [31]. We estimated the propensity scores using multinomial logistic regression with the covariates age, sex, literacy, occupation, marital status, administrative division, residence, family income, family size, living arrangement, and distance to the nearest health centre. The standardized mean difference (SMD) of 0.2 [32] among exposed and non-exposed was considered a good covariate balancing. The outcome model was the binary logistic regression on the propensity score weighted data, adjusted for potential confounders to remove the small residual covariate imbalance between the exposed and non-exposed [33].

## Ethical approval

The study protocol was approved by the institutional review board of Institute of Health Economics, University of Dhaka, Bangladesh (Ref: IHE/2020/1037). Verbal informed consent was sought from the participants before administering the survey. Participation was voluntary, and participants did not receive any compensation.

## Results

### Participants' characteristics

Table 2 describes the sociodemographic characteristics of participants, reflecting that the demographics of the study sample was predominantly aged 60–69 years (77.8%), male (65.5%), married (81.4%), devoid of formal schooling (58.3%), rural residents (73.9%), and living with family members (92.3%).

### Experienced difficulties getting medicine and receiving routine medical care

About one in four participants experienced difficulties accessing medicine (23%) and receiving routine medical care (27%) (Table 1). A significantly higher proportion of the participants with at least one condition and multimorbidity experienced difficulties accessing both medicine and receiving routine medical care (Table 2).

**Table 2. Participants' characteristics and bivariate analysis (N = 1032).**

| Characteristics | | Overall | Experienced difficulties accessing medicine (n = 240, 23.3%) | | Experienced difficulties receiving routine medical care (n = 281, 27.2%) | |
|---|---|---|---|---|---|---|
| | | n (%) | % | P | % | P |
| Administrative division | | | | | | |
| | Barishal | 149 (14.4) | 24.2 | 0.026 | 24.2 | 0.005 |
| | Chattogram | 137 (13.3) | 26.3 | | 32.1 | |
| | Dhaka | 210 (20.4) | 28.6 | | 33.8 | |
| | Mymensingh | 63 (6.1) | 17.5 | | 20.6 | |
| | Khulna | 158 (15.3) | 26.6 | | 31.7 | |
| | Rajshahi | 103 (10.0) | 22.3 | | 27.2 | |
| | Rangpur | 144 (14.0) | 12.5 | | 15.3 | |
| | Sylhet | 68 (6.6) | 20.6 | | 25.0 | |
| Age in years | | | | | | |
| | 60–69 | 803 (77.8) | 21.8 | 0.037 | 26.3 | 0.198 |
| | ≥ 70 | 229 (22.2) | 28.4 | | 30.6 | |
| Sex | | | | | | |
| | Male | 676 (65.5) | 22.3 | 0.336 | 27.4 | 0.891 |
| | Female | 356 (34.5) | 25.0 | | 27.0 | |
| Marital status | | | | | | |
| | Married | 840 (81.4) | 22.3 | 0.114 | 26.6 | 0.304 |
| | Widowed | 192 (18.6) | 27.6 | | 30.2 | |
| Formal schooling | | | | | | |
| | No | 602 (58.3) | 23.9 | 0.550 | 25.1 | 0.067 |
| | Yes | 430 (41.7) | 22.3 | | 30.2 | |
| Family size | | | | | | |
| | ≤4 | 318 (30.8) | 18.9 | 0.026 | 19.8 | <0.001 |
| | >4 | 714 (69.2) | 25.2 | | 30.5 | |
| Family income (BDT)[a] | | | | | | |
| | <5,000 | 145 (14.1) | 24.1 | 0.162 | 22.8 | <0.001 |
| | 5,000–10,000 | 331 (32.1) | 19.6 | | 16.9 | |
| | >10,000 | 556 (53.9) | 25.2 | | 34.5 | |
| Residence | | | | | | |
| | Urban | 269 (26.1) | 21.2 | 0.351 | 28.6 | 0.550 |
| | Rural | 763 (73.9) | 24.0 | | 26.7 | |
| Current occupation | | | | | | |
| | Employed | 419 (40.6) | 21.2 | 0.205 | 28.2 | 0.577 |
| | Unemployed | 613 (59.4) | 24.6 | | 26.6 | |
| Living arrangement | | | | | | |
| | Live with family | 953 (92.3) | 23.1 | 0.652 | 27.6 | 0.356 |
| | Live alone | 79 (7.7) | 25.3 | | 22.8 | |
| Walking proximity to the nearest health centre | | | | | | |
| | <30 minute | 508 (49.2) | 22.8 | 0.753 | 25.4 | 0.192 |
| | ≥30 minutes | 524 (50.8) | 23.7 | | 29.0 | |
| Non-communicable chronic conditions | | | | | | |
| | No condition | 424 (41.1) | 13.2 | <0.001 | 12.3 | <0.001 |
| | At least one condition | 372 (36.1) | 23.1 | | 30.7 | |

(*Continued*)

**Table 2.** (Continued)

| Characteristics | | Overall | Experienced difficulties accessing medicine (n = 240, 23.3%) | | Experienced difficulties receiving routine medical care (n = 281, 27.2%) | |
|---|---|---|---|---|---|---|
| | | **n (%)** | **%** | **P** | **%** | **P** |
| | Multimorbidity | 236 (22.8) | 41.5 | | 78.7 | |

[a]BDT: Bangladeshi taka. 1 BDT~0.012 United States Dollar. P-value from chi-square test comparing participants experiencing and not-experiencing difficulty.

## Prevalence of at least one condition and multimorbidity

Overall, 58.9% of study participants reported having at least one condition. Nearly one fourth (22.9%) of the participants reported having multimorbidity with a median of two conditions. Nearly a third (29.6%) of the participants had osteoarthritis, while diabetes (19.3%), hypertension (17.8%), and heart disease (12.8%) were the other most prevalent conditions (Table 3).

## Difficulties faced by the multimorbid patient for medical support during COVID-19

Table 4 shows the crude and adjusted odds ratio of having at least one condition and multimorbidity with difficulties to receive medicine and routine medical care among older Bangladeshi adults during COVID-19. On the adjusted analyses, compared to participants having no condition, those with at least one condition had two times higher odds (AOR: 1.95, 95% CI: 1.33–2.85) and those with multimorbidity had nearly five times higher odds (AOR: 4.75, 95% CI: 3.17–7.10) of experiencing difficulties accessing medicine. Similarly, in comparison to those having no condition, participants with at least one condition were thrice more likely (AOR: 3.08, 95% CI: 2.11–4.89) and those with multimorbidity were six times more likely (AOR: 6.34, 95% CI: 4.03–9.05) to face difficulties receiving routine medical care during the COVID-19 pandemic.

S1 File shows the SMD between no condition, having at least one condition and multimorbidity in unadjusted and propensity score weighted samples. Although there was an imbalance in the covariates before weighting, we observed a good balance after weighting (all SMDs ≤ 0.10). The result of sensitivity analysis using the propensity score weighting approach

**Table 3.  Prevalence of single conditions and multimorbidity (n = 1032).**

| Conditions | Prevalence (%) |
|---|---|
| Osteoarthritis | 29.6 |
| Diabetes | 19.3 |
| Hypertension | 17.8 |
| Heart disease | 12.8 |
| Chronic respiratory disease | 3.7 |
| Stroke | 3.3 |
| Hypercholesterolaemia | 2.2 |
| Chronic kidney disease | 1.7 |
| Cancer | 0.2 |
| At least one condition | 58.9 |
| Multimorbidity | 22.9 |

**Table 4. Association between multimorbidity, presence of any NCD conditions and difficulties receiving medication and health care among Bangladeshi older adults during COVID-19.**

| | Experienced difficulties accessing medicine | | Experienced difficulties receiving routine medical care | |
|---|---|---|---|---|
| | OR (95% CI) | AOR[1] (95% CI) | OR (95% CI) | AOR[1] (95% CI) |
| Non-communicable chronic conditions | | | | |
| No-condition | Ref. | Ref. | Ref. | Ref. |
| At least one condition | 1.97 (1.36–2.86) | 1.95 (1.33–2.85) | 3.16 (2.20–4.55) | 3.08 (2.11–4.89) |
| Multimorbidity | 4.67 (3.18–6.84) | 4.75 (3.17–7.10) | 6.80 (4.62–10.01) | 6.34 (4.03–9.05) |

OR: Odds ratio (unadjusted); AOR: Adjusted odds ratio; CI: Confidence interval; Ref.: Reference category.

[1] Model adjusted for age, sex, marital status, formal schooling, family size and income, residence, current occupation, living arrangement, proximity to the nearest health centre.

in the association between non-communicable chronic conditions and difficulties to receive medical support is presented in S2 File. Like the primary analysis, we observed approximately similar strength of association of non-communicable chronic conditions with facing difficulties getting medicine and facing difficulties receiving routine medical care during the COVID-19 pandemic.

## Discussion

Bangladesh has experienced an epidemiological transition, with a burgeoning older population whose social, health, and economic needs should be prioritized. Given a disproportionate number of the older population with non-communicable chronic conditions in Bangladesh [34], this study aimed to assess the challenges faced by Bangladeshi older adults with NCDs or multimorbid conditions in assessing routine health care services amid the COVID-19 pandemic.

We found that both the people with at least one condition and multimorbidity experienced significantly greater difficulties in getting medicine and receiving routine medical care than those having no condition during the COVID-19 pandemic. Two-fifth of the participants with multimorbidity experienced difficulties getting medicine during the COVID-19 pandemic. In adjusted analyses, participants with multimorbidity had three times increased odds of experiencing difficulties compared to those without multimorbidity. While our study is the first of its kind from Bangladesh conducted during the COVID-19 pandemic and lacks a previous national comparison, findings from other countries are in line with our findings [35, 36]. A recent study described the effects of the COVID-19 on the lives of people with NCDs, noting the disruption to essential public health services in Bangladesh [25].

Acknowledging that access to health care for non-communicable chronic conditions is an ongoing issue and existed in Bangladeshi society even before COVID-19 [37, 38], we believe that it is likely that the current pandemic has exacerbated this. The countrywide travel restrictions and limited availability of transportations during this pandemic have made it difficult to access healthcare centres located mainly in urban areas [39]. Notably, the most common means of transport in Bangladesh is public transportation [3], and distance to healthcare facilities and transportation might have placed significant barriers in accessing healthcare in Bangladesh [40]. The shutdown in transportation has also impacted the distribution and availability of medicine in rural areas, from where most of our sample drawn. While health insurance policy is still in a nascent stage in Bangladesh, out-of-pocket medical expenses are high in Bangladesh [41]. In light of another study reporting increased out-of-pocket cost during the COVID-19 pandemic [42], affordability of medicine and access to health care services

during the pandemic may also have contributed to the difficulty experienced by participants in our study. In Bangladeshi society, the male adult is morally obliged to provide care and support to their older parents. While many of our participants may not have been actively employed, they were nevertheless economically dependent on adult children who may have lost a source of income during the pandemic, subsequently impacting our participant's ability to pay for health care and medicine.

Additional factors such as changed priorities within the health sector to COVID-19 should also be considered as possible barriers. Many countries' healthcare systems, including Bangladesh, were strained to accommodate the increasing number of COVID-19 patients [28, 43]. Moreover, many of the tertiary care centres were converted into COVID-19 hospitals, limiting access to regular health care services [39]. Especially for older adults with multimorbidity, regular medical check-ups and access medications are important ways to avoid the severe consequences of their diseases including disability or mortality [44]. Although the broader impact of inaccessibility on health care for multimorbid patients is yet to be fully understood, national response framework and plans should be more holistic and have plan to address both emerging and existing needs of older adults with non-communicable chronic conditions, that will ensure the health and well-being during the current COVID-19 pandemic.

Acknowledging that, with immunisation, the impact of the COVID-19 pandemic on health care is likely to become less severe, this study's broader implications should be contextualized for future pandemics, public health emergencies, and disasters of national or global scales. Bangladesh's high migration, crowded urban areas, poor sanitation, and fragile health care system are likely to incubate further outbreaks, including COVID-19 variants [28]. At the same time, the increasing prevalence of NCDs and multimorbidity, the traditional "just-in-time" approach of dealing with crises, and inadequately resourced health facilities make it more vulnerable to future pandemics and resulting loss and sufferings [45, 46]. WHO defines pandemic preparedness as "having national response plans, resources, and the capacity to support operations in the event of a pandemic". During a pandemic, in addition to preventing, detecting, and containing outbreaks, health systems need to continue to deliver essential health care, especially to the most vulnerable populations like older adults with non-communicable disease. The current pandemic provides us with an opportunity to evaluate our existing practices to deliver essential health care services during emergencies to the most vulnerable groups and strengthen national emergency preparedness system. Engagement of both private and public healthcare institutions along with other stakeholders in health service design [47] would be essential design people centred care model to maintain access to basic health services (such as screening, medical check-ups, and pharmacy services) during the emergency and beyond. Decision makers should also design the *s*trategies that can address the needs of the older adults with non-communicable disease to provide basic health care services required for maintaining good health at the community level through primary health care approach. health care access that we found in Bangladesh during the COVID-19 pandemic.

Moreover, non-government organizations, health volunteers, and community health workers working locally can be important partners in ensuring the delivery of essential drugs for NCDs management, especially to the socio-economically vulnerable older people, and those lacking mobility and social support [48]. To reduce economic dependency on family members and promote autonomy, a comprehensive health and social care program need to be introduced to ensure access for older adults. This may include telehealth, remote health consultations, and dissemination of people friendly health information and resources.

## Strengths and limitations of the study

This is the first nationwide study carried out in Bangladesh, exploring the challenges faced by the patients suffering from chronic conditions in receiving routine medical care amid this COVID-19 pandemic. Also, our findings of having higher odds of facing difficulties to receive medical support among the participants with at least one non-communicable chronic condition and multimorbidity are robust to sensitivity analysis when we applied the propensity score approach. However, the study also had several limitations. Firstly, there is a possibility of selection bias as the sampling frame was developed from the available household-level information in our registry. Secondly, all the information was self-reported and based on the participants' perceptions. Moreover, we were not able to access the participants' medical records to verify their chronic conditions.

## Conclusion

We found that Bangladeshi older adults with NCDs or multimorbidity face challenges in receiving routine medical treatments during this COVID-19 pandemic. Policymakers and health care practitioners need to consider innovative strategies to ensure continued care provision and routine medical care to older people with NCDs during COVID-19 and prepare adequately for similar needs during any future public health emergencies. Future research should also focus on strengthening the fragile health care system so that it has the capacity to support chronic patients in emergencies such as the COVID-19 pandemic.

## Supporting information

**S1 Data. Data file of the study.**
(DTA)

**S1 File. Standardized mean differences (SMD) between no non-communicable chronic condition, at least one condition, and multimorbidity in unadjusted and propensity score weighted samples.**
(DOCX)

**S2 File. Sensitivity analysis using propensity score analysis for the association between multimorbidity and difficulties receiving medical support among Bangladeshi older adults during COVID-19.**
(DOCX)

## Acknowledgments

We acknowledge the role of Sadia Sumaia Chowdhury, Programme Manager, ARCED Foundation and Muntasir Alam, Research Assistant, ARCED Foundation for their support in data collection for the study.

## Author Contributions

**Conceptualization:** Sabuj Kanti Mistry, A. R. M. Mehrab Ali, Uday Narayan Yadav.

**Data curation:** Sabuj Kanti Mistry, A. R. M. Mehrab Ali.

**Formal analysis:** Sabuj Kanti Mistry, A. R. M. Mehrab Ali.

**Investigation:** Sabuj Kanti Mistry.

**Methodology:** Sabuj Kanti Mistry, A. R. M. Mehrab Ali.

**Project administration:** Sabuj Kanti Mistry, A. R. M. Mehrab Ali.

**Resources:** Sabuj Kanti Mistry.

**Software:** Sabuj Kanti Mistry.

**Supervision:** Sabuj Kanti Mistry, A. R. M. Mehrab Ali.

**Validation:** Sabuj Kanti Mistry.

**Visualization:** Sabuj Kanti Mistry.

**Writing – original draft:** Sabuj Kanti Mistry, A. R. M. Mehrab Ali, Uday Narayan Yadav, Saruna Ghimire, Md. Belal Hossain, Suvasish Das Shuvo, Manika Saha, Sneha Sarwar, Md. Mohibur Hossain Nirob.

**Writing – review & editing:** Varalakshmi Chandra Sekaran, Mark F. Harris.

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
