## [Decision Letter · Decision Letter 0]

28 Jun 2021

PONE-D-21-17040

Older adults with non-communicable chronic conditions and their health care access amid COVID-19 pandemic in Bangladesh: findings from a cross-sectional study

PLOS ONE

Dear Dr. Mistry,

Thank you for submitting your manuscript to PLOS ONE. After careful consideration, we feel that it has merit but does not fully meet PLOS ONE’s publication criteria as it currently stands. Therefore, we invite you to submit a revised version of the manuscript that addresses the points raised during the review process.

Please pay attention in English so that the paper is readable. Also improve Introduction and conclusion providing details on literature reviews, research gap and findings. 

We look forward to receiving your revised manuscript.

Kind regards,

Enamul Kabir

Academic Editor

PLOS ONE

Journal Requirements:

Reviewers' comments:

Reviewer's Responses to Questions

**Comments to the Author**

1. Is the manuscript technically sound, and do the data support the conclusions?

Reviewer #1: Yes

Reviewer #2: Yes

2. Has the statistical analysis been performed appropriately and rigorously? 

Reviewer #1: Yes

Reviewer #2: Yes

3. Have the authors made all data underlying the findings in their manuscript fully available?

Reviewer #1: Yes

Reviewer #2: Yes

4. Is the manuscript presented in an intelligible fashion and written in standard English?

Reviewer #1: Yes

Reviewer #2: Yes

5. Review Comments to the Author

Reviewer #1: The authors reported on access to health care for elderly people with NCDs in Bangladesh and revealed that elder people with NCD faced difficulties receiving routine medical care during the pandemic. It might be a serious issue for people in the world. Therefore, their findings are valuable real-world data.

However, I have small difficulty in understanding their research. The following comments maybe helpful the authors.

1. Definition of NCDs is unclear (Table 1). I was concerned about the difference between taking some medications and self-reported. Although both are inherently self-reported, in the case of "Self-reported"(not taking medication or therapy), how did the authors determine if they need to go to hospital or clinic.

2. Could you provide the average or median of multimorbidity? I am interested in that data because there is a differnce from a minimum of 2 to a maximum of 9.

Reviewer #2: Authors focused a burning and time demanding topic to discuss in this pandemic. Methodology was clearly discussed, findings were presented appropriately. Please, edit the conclusion section of abstract.

6. PLOS authors have the option to publish the peer review history of their article (what does this mean?). If published, this will include your full peer review and any attached files.

Reviewer #1: No

Reviewer #2: No

---

## [Author Response · Author response to Decision Letter 0]

2 Jul 2021

Dear Editor,

We greatly appreciate the valuable comments from the Editor and the reviewers and have modified the enclosed manuscript accordingly. Here, we include our responses to each of the reviewers’ comments.

Editor

Author’s response: We confirm that the manuscript complies with the PLOS One guidelines

Reviewer 1

1. Definition of NCDs is unclear (Table 1). I was concerned about the difference between taking some medications and self-reported. Although both are inherently self-reported, in the case of "Self-reported"(not taking medication or therapy), how did the authors determine if they need to go to hospital or clinic.

Author’s response: Thank you very much for bringing this to our notice, and we apologize for the ambiguity. We have revised our statements for clarity by adding the questions that were asked to the participants. Our revised paragraph reads as below on page 7 line 171-179 of the revised manuscript. 

“The data on nine non-communicable chronic conditions (Hypertension, Heart diseases, Stroke, Hypercholesterolemia, Diabetes, Chronic respiratory diseases, Chronic kidney disease, Cancer, and Osteoarthritis) were collected through participants’ self-reports. For each of the conditions, participants were asked two questions: 1) “Has a doctor or other health professional ever told you that you have [the condition]?” and 2) “Are you currently taking any medications for [the condition]?” Participants were classified as having a given condition if they reported yes to either of the two questions. The definitions of the individual non-communicable condition, having at least one condition and multimorbidity are presented in table 1.” 

2. Could you provide the average or median of multimorbidity? I am interested in that data because there is a differnce from a minimum of 2 to a maximum of 9.

Author’s response: Thank you. The median of multimorbidity was 2 which have been added in the in the revised manuscript (Result section under sub-heading “Prevalence of at least one condition and multimorbidity”). Please see page 12 line 246-248.

 

Reviewer 2

Authors focused a burning and time demanding topic to discuss in this pandemic. Methodology was clearly discussed, findings were presented appropriately. Please, edit the conclusion section of abstract.

Author’s response: Thank you very much for appreciating our paper. We have edited the conclusion of the abstract as suggested. 

Concluding Remarks

We are extremely appreciative of the Editor’s and reviewers’ time and helpful comments and hope that our revisions have adequately addressed their concerns. We are confident that the revisions have strengthened our manuscript. We look forward to hearing from you with a final decision regarding the acceptance of our manuscript.

Regards,

Authors

---

## [Decision Letter · Decision Letter 1]

19 Jul 2021

Older adults with non-communicable chronic conditions and their health care access amid COVID-19 pandemic in Bangladesh: findings from a cross-sectional study

PONE-D-21-17040R1

Dear Dr. Mistry,

We’re pleased to inform you that your manuscript has been judged scientifically suitable for publication and will be formally accepted for publication once it meets all outstanding technical requirements.

Kind regards,

Enamul Kabir

Academic Editor

PLOS ONE

Additional Editor Comments (optional):

Reviewers' comments:

Reviewer's Responses to Questions

**Comments to the Author**

1. If the authors have adequately addressed your comments raised in a previous round of review and you feel that this manuscript is now acceptable for publication, you may indicate that here to bypass the “Comments to the Author” section, enter your conflict of interest statement in the “Confidential to Editor” section, and submit your "Accept" recommendation.

Reviewer #1: All comments have been addressed

Reviewer #2: All comments have been addressed

2. Is the manuscript technically sound, and do the data support the conclusions?

Reviewer #1: Yes

Reviewer #2: Yes

3. Has the statistical analysis been performed appropriately and rigorously? 

Reviewer #1: Yes

Reviewer #2: Yes

4. Have the authors made all data underlying the findings in their manuscript fully available?

Reviewer #1: Yes

Reviewer #2: Yes

5. Is the manuscript presented in an intelligible fashion and written in standard English?

Reviewer #1: Yes

Reviewer #2: Yes

6. Review Comments to the Author

Reviewer #1: Thank you for your great work. The authors has strengthened the manuscript. I am confident that this manuscript can contribute to the health policies of Bangladesh.

Reviewer #2: Current study focused on a time demanding issue. Authors did a good review of existing literature, methodology and results were clear.

7. PLOS authors have the option to publish the peer review history of their article (what does this mean?). If published, this will include your full peer review and any attached files.

Reviewer #1: No

Reviewer #2: No

---

## [Editor Report · Acceptance letter]

22 Jul 2021

PONE-D-21-17040R1 

Older adults with non-communicable chronic conditions and their health care access amid COVID-19 pandemic in Bangladesh: findings from a cross-sectional study 

Dear Dr. Mistry:

I'm pleased to inform you that your manuscript has been deemed suitable for publication in PLOS ONE. Congratulations! Your manuscript is now with our production department. 

Kind regards, 

on behalf of

Dr. Enamul Kabir 

Academic Editor

PLOS ONE